# Peer review of "Advanced Molecular Dynamics Approaches to Model a Tertiary Complex APRIL/TACI with Long Glycosaminoglycans"

_biomolecules, 2021, doi:10.3390/biom11091349_

Round 1

Reviewer 1 Report

Authors have performed a docking study of 24 and 48 mer Glycosaminoglycans (GAGs) with APRIL/TACI complex. They have employed RS-REMD, a replica-exchange based scheme employing different levels of repulsive biasing between partners in each replica simulation. The bias acts only on intermolecular interactions based on an increase in effective pairwise van der Waals radii (repulsive scaling (RS)-REMD) without affecting interactions within each protein or with the solvent. In this work, authors have used 24 and 48 mer GAG compared to their earlier work with 4 and 6mer (IJMS, 2019). This work is interesting as it involves examining the interaction of larger units of GAG and a large protein complex.

  1. In the abstract, the following sentence can be rewritten:

“We built a model of long GAGs in complex with a proliferation-inducing ligand pre- 20 bound to its receptors, the B cell maturation antigen and the transmembrane activator and calcium 21 modulator and cyclophilin ligand interactor and evaluated prediction power of the RS-REMD for 22 this tertiary complex.”

  1. Two long tables, Table 1 and Table 2, can be moved to SI, and significant findings can be described with a couple of graphical representations summarizing these two tables. It would more readily get the vital point across powerfully.

Author Response

Attached as a PDF file.

Reviewer 2 Report

The manuscript presents a study of glycosaminoglycan oligosaccharides' conformational preferences and their impact on binding and ternary complex formation. This is a very timely and relevant work given how under-investigated the sugars as a major class of biomolecules compared to the lipids, proteins, and nucleic acids. Due to conformational flexibility, polysaccharides present unique challenges to sampling in detailed atomistic simulations. Authors have employed state of the art to enhance sampling to obtain the representative states of conformational ensembles.

I have a few components to help improve the manuscript. One of the major criticism is the absence of quantitative statistical measures to characterize conformational ensembles, e.g, distributions of dihedral angles, Rg, RMSD, contact maps, etc. Such analysis would reveal additional insights and help readers appreciate the degree of conformational flexibility of these systems. See papers by J Xia et al J. Am. Chem. Soc. 2011, 133, 15252–15255 , J Roche et al J Phys Chem B, 2019 123, 9567-9575 and Sairam S. Mallajosyula et al Methods Mol Biol. 2015 ; 1273: 407–429 

  1. Authors, please expand the acronyms in the abstract and everywhere in the text. APRIL/TACI is mentioned in the abstract without clear expansion.
  2.  Structures of energetically most stable conformations are shown on Fig 1 and 2 but it would be very useful also to show Rg, RMSD distributions coming from MD to get a sense of conformational diversity, heterogeneity. Alongside fig 2 and Fig 3, authors may also want to plot some measure of contact number distribution for binding partners.
  3. Could authors plot the energies and some structural measure of docking in a 2D plot comparing REMD to Autodock3 to more forcefully drive home one of their central claims? "In this study, RS-REMD showed clear superiority over Autodock3 in case of 270 docking quality for such long molecules."

Author Response

Attached as a PDF file.

Reviewer 3 Report

The authors of manuscript use MD-based methods to describe interaction of GAG polymers with APRIL/TACI and APRIL/BCMA protein complexes. The problem is scientifically sound and challenging. The authors use conventional docking and RS-REMD simulations to dock long GAG’s which is followed by the refinement procedure. I have following major issues related to the methodology applied to these systems:

  • One of the conclusions of authors states that HP dp48 interacts with positively charged residues located on TACI receptor which causes bending of HP dp48 molecule. This bending is visible in Figure 3 (bottom). IMO visual comparison with the same structures after refinement shown in Figure 4 suggests that this bending is (slightly?) released after refinement. It is difficult to judge the degree of relaxation just from visual inspection, but more importantly it is difficult to judge from presented data if the system actually reached the equilibrium (no errors of dG’s are presented). Authors state that the refinement procedure was 25 ns long. There is no proof that this relatively short simulation is sufficient for the system to reach equilibrium. I suggest running much longer simulations and checking whether reported observables (like free energies from MM-GBSA method) really converged. It can be done by dividing the trajectory into several intervals and calculating desired observables in these “windows”. It should be done for all investigated systems. This way also uncertainties of determined observables (like dG’s) can be estimated and authors could answer the question whether long GAGs really interact with the receptors.
  • Another methodological concern is related to the application of 1-trajectory MM-GBSA methodology to the investigated systems. GAG’s seems to be very flexible molecules and therefore the negative conformational entropy change upon binding might be very significant for this system. This entropy change might be overcame by enthalpic contribution of charge-charge interactions, but it is not certain if it is the case from the presented data, as classical 1-trajectory MM-GBSA estimates mostly enthalpic part of the free energy. IMO reliable method of estimation of conformational entropy change upon binding is necessary for presented systems. Perhaps authors should consider running 3-trajectory MM-GBSA with NMA for entropy estimation or using different method of entropy change estimation (see Panday and Gosh JCTC 2020 16: 7581).
  • Description of the methodology should be significantly improved.

To summarize. IMO the conclusion that the TACI-GAG interaction can be attenuated (or amplified?) by APRIL cannot be drawn from presented data because: 1. There is no proof that actual simulations reached equilibrium (converged). 2. The conformational entropy change of GAGs upon binding is not evaluated. These problems should be fixed.

Other problems with the manuscript:

  • abbreviations TACI and APRIL used in abstract should be explained there
  • line 33 - wouldn't be "negative charge distribution" more appropriate than "charge densities"?
  • lines 74-75 - authors should provide reference here
  • lines 94-97 - authors describe starting structures of proteins, but what are the starting structures of GAG's in RS? Extended, random? It should be described here.
  • lines 96-97 - description of FF parameters is very enigmatic. If not pure GLYCAM06 force-field was used it should be precisely explained what kind of modifications were applied to this FF. Moreover on line 110 only GLYCAM06 FF is mentioned without mention of reference 38. 
  • line 109 - "The ff14SBonlysc force field parameters41" but the ref 41 points to ff99SB force field. So which FF was used?
  • lines 113-125 - does this section pertain to RS-REMD simulations or to the refinement of AD3 structures? or both? What water model was used? The harmonic restraints were placed on which atoms? all, heavy, backbone? There should be references to both PME and SHAKE. What was the timestep and cutoffs for electrostatics? The second sentence in this paragraph is also not clear and should be improved. What were the protonation states of titrable groups?
  • lines 129-130 - was LIE analysis really used in this work? Where in the manuscript LIE analysis was used?
  • lines 135-136 - it should be stated clearly how many central glycosydic linkages were kept flexible and why this number was chosen. It is also not clear whether only glycosydic linkages were flexible. Were central whole sugars flexible or rigid? It should be clearly described.
  • line 190 - why only electrostatic component of energy not free energy from MM-GBSA was used for preselection?
  • Figure 3 bottom - should be APRIL-TACI-HP dp48 not dp24
  • Figure 4 - is there any reason of changing color of representation of APRIL? In my opinion it is better for the reader if it is kept the same as in figure 3.
  • line 228 - should be APRIL-receptor
  • lines 253-254 - I think it is better to show average free energies not minimal to show the trend
  • line 255-256 - "All these results suggest that in case a long GAG molecule is indeed able to bind to both APRIL and its receptors attenuating their interactions" - attenuating interactions of APRIL with receptors or GAG-APRIL or GAG-receptor. It is not clear from this sentence.
  • lines 268-270 - from this sentence it is clear that authors meant attenuation of TACI-GAG interactions. I don't quite understand why authors wrote about attenuation if they state that TACI-GAG interactions without meditation of APRIL were not observed (line 267). Isn't it rather amplification not attenuation of TACI-GAG interactions? 
  • Table 1 - why one value of dG is missing?

Author Response

Attached as a PDF file.

Round 2

Reviewer 2 Report

The authors have sufficiently revised the manuscript. I have no further comments. 

Reviewer 3 Report

The manuscript has been sufficiently improved and can be published.